# G9a regulates temporal preimplantation developmental program and lineage segregation in blastocyst

Jan J Zylicz[1,2,3†], Maud Borensztein[1,2†], Frederick CK Wong[1,2], Yun Huang[1,2], Caroline Lee[1,2], Sabine Dietmann[3], M Azim Surani[1,2]*

[1]Wellcome Trust/Cancer Research United Kingdom Gurdon Institute, University of Cambridge, Cambridge, United Kingdom; [2]Department of Physiology, Development and Neuroscience, University of Cambridge, Cambridge, United Kingdom; [3]Wellcome Trust/Medical Research Council Stem Cell Institute, University of Cambridge, Cambridge, United Kingdom

**Abstract** Early mouse development is regulated and accompanied by dynamic changes in chromatin modifications, including G9a-mediated histone H3 lysine 9 dimethylation (H3K9me2). Previously, we provided insights into its role in post-implantation development (Zylicz et al., 2015). Here we explore the impact of depleting the maternally inherited G9a in oocytes on development shortly after fertilisation. We show that G9a accumulates typically at 4 to 8 cell stage to promote timely repression of a subset of 4 cell stage-specific genes. Loss of maternal inheritance of G9a disrupts the gene regulatory network resulting in developmental delay and destabilisation of inner cell mass lineages by the late blastocyst stage. Our results indicate a vital role of this maternally inherited epigenetic regulator in creating conducive conditions for developmental progression and on cell fate choices.

DOI: https://doi.org/10.7554/eLife.33361.001

[†]These authors contributed equally to this work

**Competing interests:** The authors declare that no competing interests exist.

## Introduction

The first developmental events in mouse are subject to regulation by information stored in the oocyte. The maternal inheritance in oocytes consists of mRNAs and proteins, which together direct rapid epigenetic reprogramming upon fertilisation, cell cleavage and activation of the zygotic genome (*Ancelin et al., 2016*; *Li et al., 2010*; *Wasson et al., 2016*).

The process of maternal-to-zygotic transition (MZT) in mice is first initiated in the late zygote, becoming more prominent at 2 cell stage (2C) on embryonic day (E) 1.5 (*Golbus et al., 1973*; *Hamatani et al., 2004*; *Peaston et al., 2004*). The initial wave of activation of many genes is followed by their repression within one or two cell cycles (*Falco et al., 2007*; *Hamatani et al., 2004*). How multiple genes are faithfully repressed during early embryogenesis to promote developmental progression remains unclear.

Extensive remodelling of the histone modifications and DNA methylation accompany division of blastomeres, which suggests a role for epigenetic regulatory mechanisms during development of the first embryonic lineages (*Dahl et al., 2016*; *Liu et al., 2016*; *Smith et al., 2012*; *Wang et al., 2014*; *Wu et al., 2016*; *Zhang et al., 2016*; *Zheng et al., 2016*). Outer trophectoderm (TE) cells are the first to be allocated together with the inner cell mass cells (ICM) at the blastocyst stage (E3.5). Shortly afterwards, by E4.0, the ICM segregates into primitive endoderm cells (PrE) and the pluripotent pre-Epiblast cells (Epi), which will give rise to the yolk sac and embryo proper respectively (*Chazaud et al., 2006*; *Schrode et al., 2013*).

We extend our previous study on the role of G9a-mediated H3K9me2 in mouse early post-implantation development (*Zylicz et al., 2015*), and examined the role of this histone methyltransferase during preimplantation development. We show that G9a (encoded by *Ehmt2* gene) is maternally inherited and drives the accumulation of H3K9me2 at 4C and 8C stage, which accounts for timely repression of a subset of transcripts expressed at 4C. Severe disruption of the gene regulatory network follows upon maternal loss of G9a, resulting in developmental delay and destabilisation of ICM lineages, and frequent loss of embryos at the peri-implantation stage. Altogether, our results indicate that maternally-inherited G9a is crucial for regulating appropriate gene expression changes during preimplantation development.

## Results and discussion

### G9a and H3K9me2 accumulate at 4 and 8 cell stage

First, we investigated the H3K9me2 dynamics in early mouse preimplantation development. Immunofluorescence (IF) analysis of 2C (E1.5), 4C (E2.0), 8C (E2.5) and late blastocysts (E4.5) revealed progressive and significant accumulation of H3K9me2 at 4C and 8C stage; this was not the case at 2C or at E4.5 (*Figure 1A,B*). A more substantial enrichment follows in the epiblast of postimplantation embryos (*Zylicz et al., 2015*).

The first wave of H3K9 dimethylation is consistent with increased level of G9a at 4C, and more significantly so at 8C stage (*Figure 1A,B*). What is more, G9a's binding partner GLP also accumulates in 8C embryos (*Figure 1—figure supplement 1*). These results, in line with a previous immunofluorescence study (*Li et al., 2013*), indicate that even low levels of nuclear G9a at 4C stage are sufficient to initiate H3K9 dimethylation. Thus, following the burst of transcription at 2C stage, blastomeres accumulate substantial levels of repressive H3K9me2 mark, although its functional significance remains unclear.

### G9a promotes developmental progression and primitive endoderm (PrE) segregation

To determine the role of G9a in preimplantation development, we first induced depletion of the maternal pool of both the *Ehmt2* transcript and G9a protein in the oocyte (*Figure 1—figure supplement 2*). To do so, we used *Zp3-Cre*-expressing *Ehmt2$^{F/-}$* females (*Ehmt2$^{F/+}$* for controls), in which conditional allele is recombined under Cre recombinase expression during oogenesis (*de Vries et al., 2000*), and crossed them with *Ehmt2$^{+/-}$* males. Whereas loss of both maternal and zygotic G9a (*Ehmt2$^{M/Z}$*) does not appear to grossly affect early development up to the E2.5 stage, these embryos show slight developmental delay and lack of substantial accumulation of H3K9me2 (*Figure 2A,B*, *Figure 2—figure supplement 1A*). Upon further development in vitro, the maternally depleted embryos do form blastocysts (E4.5), however with fewer *Pou5f1*-positive ICM cells (*Figure 2—figure supplement 1B,C*).

Next, we performed a detailed IF analysis of mutant and control embryos recovered at E4.5 for SOX2, SOX17 and CDX2, the critical markers of epiblast (Epi), primitive endoderm (PrE) and trophectoderm (TE), respectively (*Figure 2C*). We found no significant difference concerning lineage allocation between *Ehmt2$^{M/Z}$* and *Ehmt2$^{M/+}$* embryos and thus decided to group them together as maternally depleted embryos (*Ehmt2$^{Mat}$*). Similarly, control embryos (*Ehmt2$^{Cntr}$*) of different genotypes (e.g. *Ehmt2$^{+/-}$* vs *Ehmt2$^{-/-}$*, data not shown) also do not reveal detectable phenotypic differences at E4.5. In contrast, *Ehmt2$^{Mat}$* compared to *Ehmt2$^{Cntr}$* blastocysts show significantly reduced cell numbers (80 vs 166 cells, respectively), and with fewer ICM (SOX2$^{+ve}$ or SOX17$^{+ve}$) cells (25.0% vs 39.1% respectively) in line with our initial observation (*Figure 2D,E*, *Figure 2—figure supplement 1B,C*). We attribute the changes primarily to a relative reduction in PrE cells (7.9% vs 25.3%), as well as to a relative increase in the ICM cells that do not display expression of any of the three lineage markers (*Figure 2F*).

This observation suggests that maternal loss of G9a results in delayed development and lineage segregation within the ICM. The observed phenotype is however not entirely due to a developmental delay since we also see defects in lineage stabilisation within the ICM, potentially resulting in cell

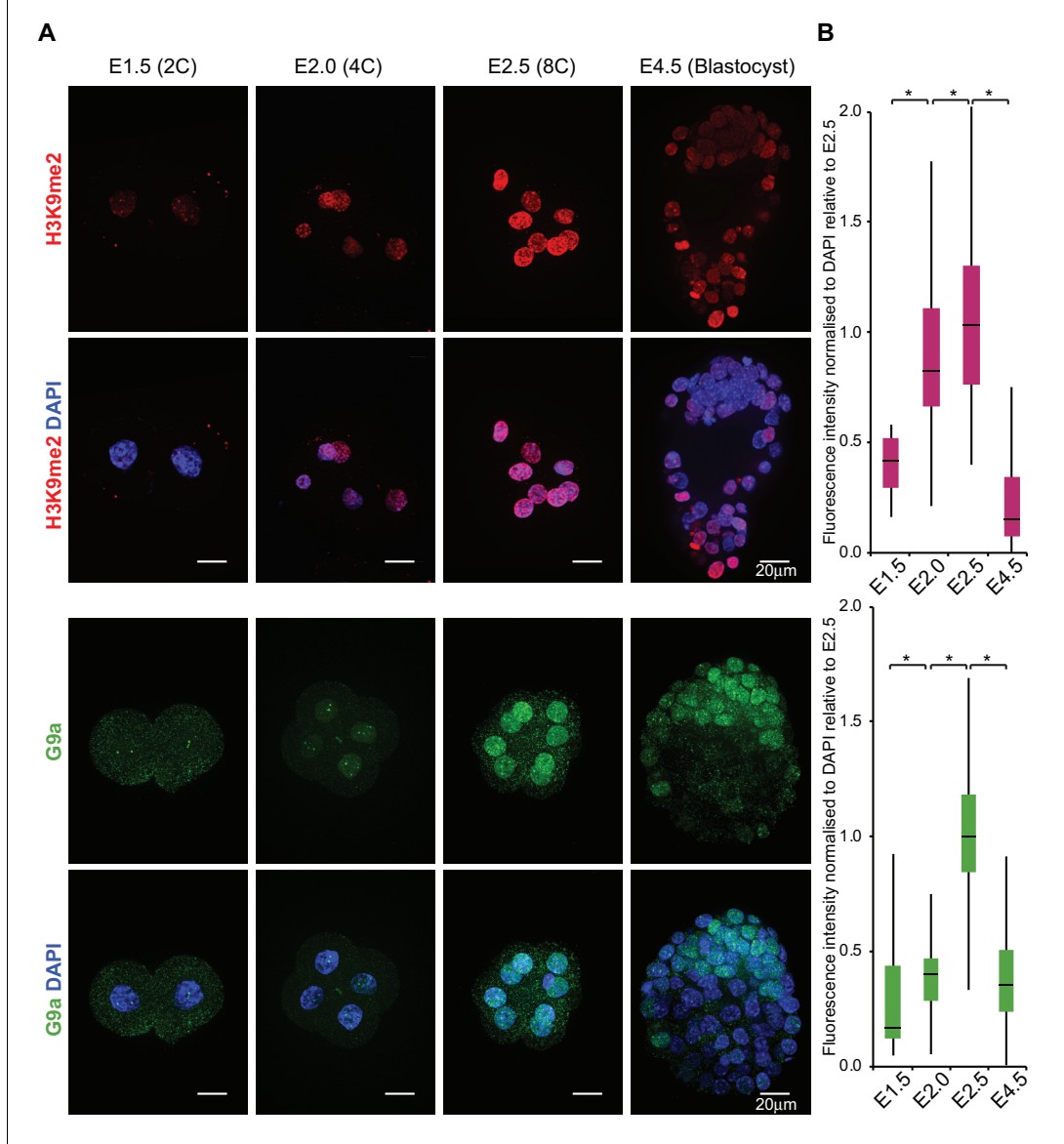

**Figure 1.** H3K9me2 and G9a accumulate at 4- and 8 cell stage. (**A**) Whole-mount IF staining for H3K9me2 (top panels) and G9a (bottom panels) in E1.5, E2.0, E2.5 and E4.5 embryos. DAPI intensity has been adjusted between time points for visualisation purposes (scale bar = 20 μm). IF signal is quantified (**B**) and visualised using box plots of median and interquartile range (IQR), with whiskers drawn 1.5xIQR away from the lower and upper quartiles. Data shows IF intensity normalised to DAPI for individual cells. At least 9 embryos were quantified for each time point. (*p<0.05 by Wilcoxon rank sum test). 2C: 2 cell stage; 4C: 4 cell stage; 8C: 8 cell stage; DAPI: 4',6-diamidino-2-phenylindole; H3K9me2: histone H3 lysine 9 dimethylation; IF: immunofluorescence; IQR: interquartile range. Also see *Figure 1—figure supplements 1* and *2*.
DOI: https://doi.org/10.7554/eLife.33361.002

The following figure supplements are available for figure 1:

**Figure supplement 1.** GLP accumulates at 8C (E2.5) stage.
DOI: https://doi.org/10.7554/eLife.33361.003

**Figure supplement 2.** Both G9a and GLP are maternally inherited.
DOI: https://doi.org/10.7554/eLife.33361.004

loss. Consistently, when comparing *Ehmt2*<sup>Mat</sup> and *Ehmt2*<sup>Cntr</sup> blastocysts we observed a significant increase in the levels γH2A.X, a hallmark of genetic stress and misregulation of the cell cycle (*Figure 2—figure supplement 1B,C*) (*Turinetto and Giachino, 2015*). Accumulation of γH2A.X also occurs in cells with decondensed chromatin potentially due to loss of H3K9me2 (*Banáth et al.,*

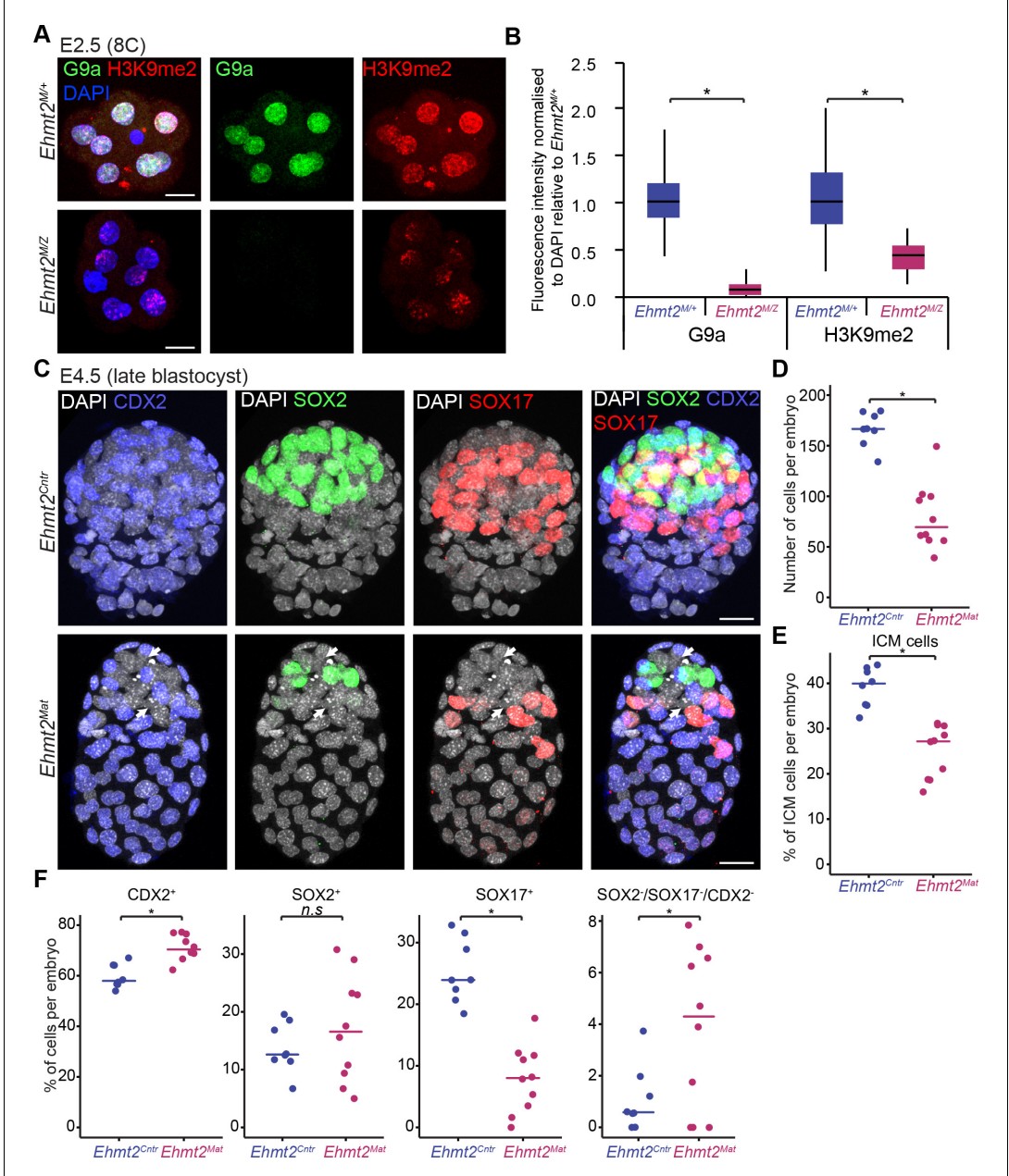

**Figure 2.** Lack of maternal G9a leads to smaller blastocysts with fewer PrE cells. (A) Whole-mount IF staining for G9a and H3K9me2 in E2.5 *Ehmt2^{M/Z}* and *Ehmt2^{M/+}* embryos. (Scale bar = 20 μm). (B) IF signal quantification for G9a and H3K9me2 from *Figure 2A*. Box plots show median and interquartile range (IQR), with whiskers drawn 1.5xIQR away from the lower and upper quartiles. Data shows IF intensity normalised to DAPI. At least six embryos were quantified for each genotype. (*p<0.05 in Wilcoxon rank sum test). (C) Whole-mount IF staining of E4.5 *Ehmt2^{Mat}* and *Ehmt2^{Cntr}* blastocysts using anti-CDX2 (TE, Blue), anti-SOX2 (Epi, Green) and anti-SOX17 (PrE, Red) antibodies. White arrows point towards nuclei devoid of staining for any lineage marker. (Scale bar = 20 μm). (D–F) Dot plots showing IF quantification from *Figure 2C* in relation to embryo genotypes. Each dot represents one embryo, 10 *Ehmt2^{Mat}* and 8 *Ehmt2^{Cntr}* embryos were quantified. (D) Total number of cells in an embryo. (E) Percentage of cells within the ICM (SOX2^+ or SOX17^+). (F) Percentage of cells within each lineage or showing no marker gene expression (SOX2^-SOX17^-CDX2^-). Line shows the median. (*p<0.05 in Wilcoxon rank sum test). CDX2: Caudal Type Homeobox 2; DAPI: 4',6-diamidino-2-phenylindole; *Ehmt2^{Cntr}*: control embryos with maternally inherited G9a; *Ehmt2^{M/+}* embryos maternally depleted but with zygotic expression of G9a; *Ehmt2^{M/Z}* embryos without both maternal and zygotic expression of G9a; *Ehmt2^{Mat}* embryos maternally depleted of G9a;Epi: pre-epiblast; H3K9me2: histone H3 lysine 9 dimethylation; ICM: inner cell mass; IF: immunofluorescence; IQR: interquartile range; PrE: primitive endoderm; SOX2: SRY box 2; SOX17: SRY box 17; TE: trophectoderm. Also see *Figure 2—figure supplement 1*.

DOI: https://doi.org/10.7554/eLife.33361.005

*Figure 2 continued on next page*

*Figure 2 continued*

The following figure supplement is available for figure 2:

**Figure supplement 1.** Loss of maternal G9a results in developmental delay and increased γH2A.X staining with fewer ICM cells.

DOI: https://doi.org/10.7554/eLife.33361.006

*2009*). Notably, 'knockdown' of GLP, a binding partner of G9a, also results in defects in blastocyst, increased cell death, and loss of cells in the ICM (*Huang et al., 2015*). Thus, G9a-GLP complex likely promotes developmental progression in vivo, with a timely specification of the PrE and stabilisation of lineage choices. The significant reduction in the number of cells in mutant embryos as well as accumulation of γH2A.X indicate potential misregulation of the cell cycle as we reported in the post-implantation epiblast (*Zylicz et al., 2015*). Nevertheless, mutant embryos show defects in timely lineage segregation and stabilisation at the blastocyst stage, which indicates that epigenetic programming might create conditions conducive to cell fate choices. Previous studies have similarly indicated that arginine methylation by CARM1 and PRDM14, might also be important for the establishment of the Epi (*Burton et al., 2013*; *Torres-Padilla et al., 2007*). We thus hypothesise that maternal G9a is involved in setting up a stable transcriptional and epigenetic network allowing for timely developmental progression and lineage segregation at the blastocyst stage.

## G9a represses a subset of 4C upregulated genes

To address the functional relevance of G9a in transcriptional regulation, we focused on the 8C stage (E2.5), when we detect accumulation of the highest levels of G9a and H3K9me2. Loss of maternal G9a at 8C has yet to result in an overt phenotype except for a mild developmental delay. To determine if there are underlying consequences of loss of G9a already at 8C, we performed single-embryo RNA-seq on ten control ($Ehmt2^{Cntr}$), and ten maternally depleted embryos ($Ehmt2^{Mat}$), which were morphologically indistinguishable (*Figure 3—source data 1*). Of note, we observed differential expression of only six genes between $Ehmt2^{M/Z}$ and $Ehmt2^{M/+}$; we decided therefore to group them together. Maternally depleted G9a embryos however are largely transcriptionally distinct from the controls as shown by principal component analysis (PCA) (*Figure 3—figure supplement 1A*). The differential expression of 1467 genes accounts for the differences between the two groups, of which 44% become upregulated (*Figure 3A*, *Figure 3—source data 2*). Lack of enrichment for derepressed genes upon maternal loss of a transcriptional repressor implies that the transcriptome is affected prior to 8C, by which stage we observe substantial accumulation of secondary transcriptional effects. Interestingly, there were only four genes upregulated in both E2.5 $Ehmt2^{Mat}$ and E6.25 $Ehmt2^{-/-}$ embryos (*Zylicz et al., 2015*)(Chi2 p>0.77; *Figure 3B*), indicating that G9a plays a distinct role in pre- and post-implantation development.

For further insight into the role of G9a in regulating early development, we performed Gene Ontology (GO) enrichment analysis (*Figure 3—source data 3*). Consistent with the reduced blastocyst size and loss of ICM cells at E4.5, the GO terms indicated down regulation of genes involved in cell cycle ($p<10^{-8}$), transcription ($p<10^{-4}$) and stem cell population maintenance ($p<10^{-4}$), in $Ehmt2^{Mat}$ 8C embryos (*Figure 3—figure supplement 1B*). However, the GO term most strikingly enriched was chromatin silencing ($p<10^{-11}$), which is entirely due to reduced levels of 18 transcripts coding for H2A, MacroH2A and H2A.J (*Figure 3—figure supplement 1B,C*). Invariably, these histones are already upregulated in a subset of $Ehmt2^{Cntr}$ embryos by E2.5, which is not the case in the mutants (*Figure 3—figure supplement 1D,E*)(*Wu et al., 2014*). Thus, despite being morphologically indistinguishable from controls, $Ehmt2^{Mat}$ 8C embryos deviate from them concerning developmental progression. On the other hand, there was high enrichment of genes for GO terms such as rRNA processing ($p<10^{-23}$) and mRNA processing ($p<10^{-17}$) indicating that G9a preferentially represses genes linked to RNA metabolism (*Figure 3—figure supplement 1F*).

To put these results in the context of the developmental progression, we have integrated our results with an independent single-cell RNAseq dataset of wildtype mouse preimplantation embryos (*Tang et al., 2011*). Firstly, we performed clustering of genes using Gaussian mixture fitting based on their dynamic expression between 2C and 8C stages (*Wang et al., 2012*). To describe the expression dynamics, we found an optimal number of 14 clusters of genes. Next, we overlapped genes within clusters with those upregulated or downregulated at the 8C stage in $Ehmt2^{Mat}$

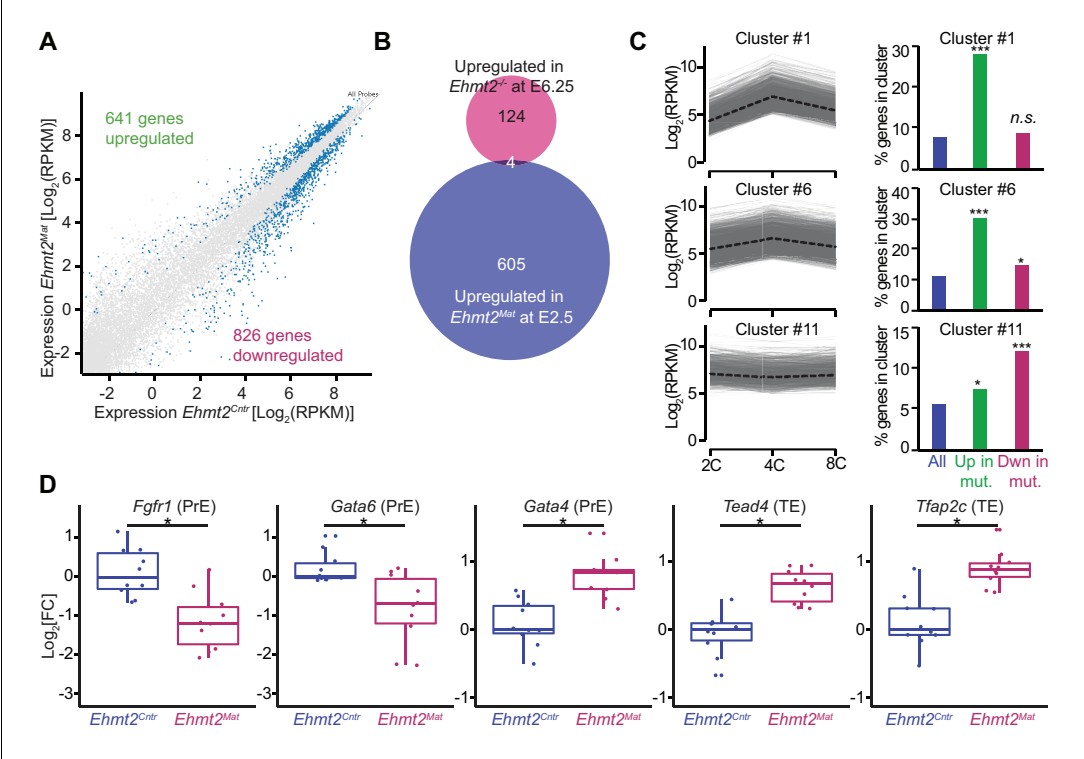

**Figure 3.** Maternal G9a represses a subset of genes induced at 4 cell stage. (**A**) Scatter plot showing transcript expression levels in *Ehmt2*$^{Cntr}$ and *Ehmt2*$^{Mat}$ 8C (E2.5) stage embryos. Blue points are differentially expressed genes (adjusted p<0.05 in DEseq2). Shown is the average from ten biological replicates. (**B**) Venn diagram showing the overlap between upregulated genes upon maternal loss of G9a (*Ehmt2*$^{Mat}$) at 8C (E2.5) and those upregulated in zygotic *Ehmt2* deletion in E6.25 epiblast (**Zylicz et al., 2015**). (**C**) Expression profiles of genes within specific clusters during wildtype development from 2C to 8C (left panels). Dotted line represents mean expression within the cluster. Data from GSE22182 (**Tang et al., 2011**) was used and within all expressed genes 14 specific clusters were identified. Shown are clusters most enriched for genes becoming derepressed (Clusters 1 and 6) or downregulated (Cluster 11) at 8C in Ehmt2$^{Mat}$. Right panel of bar plots shows the percentage of genes upregulated (Green) or downregulated (Red) at 8C in Ehmt2$^{Mat}$, which belong to identified clusters. Significance of enrichments was calculated using Chi2 test (* $10^{-5}$ < p < 0.05; ** $10^{-10}$ < p < $10^{-5}$; ***p<$10^{-10}$). (**D**) Fold expression changes of PrE (*Fgfr1, Gata6, Gata4*) and TE (*Tead4, Tfap2c*) marker genes in *Ehmt2*$^{Mat}$ compared to *Ehmt2*$^{Cntr}$ 8C embryos. Box plots show median and interquartile range (IQR), with whiskers drawn 1.5xIQR away from the lower and upper quartiles. Each dot is a single embryo and data was normalised to median expression levels in *Ehmt2*$^{Cntr}$. (*adjusted p<0.05 in DEseq2). 2C: 2 cell embryo, 4C: 4 cell embryo; 8C: 8 cell embryo; *Ehmt2*$^{Cntr}$: control embryos with maternally inherited G9a; *Ehmt2*$^{Mat}$ embryos maternally depleted of G9a; *Fgfr1*: fibroblast growth factor receptor 1; *Gata4*: GATA binding protein 4; *Gata6*: GATA binding protein 6; IQR: interquartile range; *Tcfap2c*: Transcription factor AP-2 gamma; TE: trophectoderm; *Tead4*: TEA domain transcription factor 4. Also see **Figure 3—figure supplements 1–3** and **Figure 3—source data 1–3**.

DOI: https://doi.org/10.7554/eLife.33361.007

The following source data and figure supplements are available for figure 3:

**Source data 1.** List of all single-embryo RNAseq samples sequenced.
DOI: https://doi.org/10.7554/eLife.33361.011
**Source data 2.** List of differentially expressed genes between Ehmt2$^{Cntr}$ and Ehmt2$^{Mat}$ at 8C stage embryos.
DOI: https://doi.org/10.7554/eLife.33361.012
**Source data 3.** List of enriched GO terms in genes upregulated or downregulated in 8C Ehmt2$^{Mat}$ embryos.
DOI: https://doi.org/10.7554/eLife.33361.013
**Figure supplement 1.** G9a regulates transcription of specific gene sets.
DOI: https://doi.org/10.7554/eLife.33361.008
**Figure supplement 2.** G9a represses a subset of genes upregulated at 4 cell stage.
DOI: https://doi.org/10.7554/eLife.33361.009
**Figure supplement 3.** Maternal G9a is dispensable for TE repression.
DOI: https://doi.org/10.7554/eLife.33361.010

embryos (*Figure 3C*). Strikingly, of the upregulated genes, 28% and 30% were part of cluster 1 and 6, respectively (Chi$^2$ <10$^{-68}$ and Chi$^2$ <10$^{-48}$). Genes within those clusters are transiently upregulated at the 4C stage. On the other hand, cluster 11 of stably expressed genes, is most enriched for genes downregulated upon loss of maternal G9a (12% of genes, Chi2 <10$^{-15}$, *Figure 3C*). To further validate the specific derepression of 4C genes in mutants, we have also analysed gene expression changes between 2C and 4C, as well as 4C and 8C stages in wildtype embryos (*Figure 3—figure supplement 2A*). Our analysis revealed that genes upregulated in *Ehmt2$^{Mat}$* 8C embryos (green box), show a significant increase in expression at 4C stage, which are silenced later in wildtype embryos. Our findings indicate that during preimplantation development, G9a mediates silencing of a specific set of genes, which are transiently upregulated at 4C stages. Whereas there is significant accumulation of H3K9me2 globally as judged by IF analysis at the 4C stage (*Figure 1A,B*), the precise distribution of the mark within the genome cannot be ascertained by this observation. We propose that it is only at the 8C stage that the increasing levels of G9a levels allow H3K9me2 deposition at the 4C-specific genes.

Next, to explore the roots of lineage destabilisation in maternally depleted embryos, we focused on the misexpression of transcriptional factors (TFs) in our RNAseq dataset (n = 71) (*Figure 3—figure supplement 2B*). Consistent with fewer PrE and more TE cells, there was a reduction in the levels of PrE specifiers (*Gata6, Fgfr1*), and an increase in the expression of TE markers (*Tead4, Tcfap2c*) in maternally deleted mutants compared to controls (*Figure 3D*). Intriguingly, expression of *Gata4*, normally seen in late blastocysts, already shows upregulation in mutant embryos (*Plusa et al., 2008*) (*Figure 3D*). These results indicate that maternal loss of G9a results in destabilisation of specific gene regulatory networks involved in pre-implantation development. Efficient setting up of such transcriptional circuitry as early as the 8C stage seems necessary for subsequent timely lineage segregation. However, it remains unclear which subset of the differentially expressed genes directly results in PrE defects. Indeed, this might be linked to increased expression of a critical trophectoderm specifier *Tead4* (*Yagi et al., 2007*), or downregulation of vital PrE regulators *Fgfr1* (*Kang et al., 2017*; *Molotkov et al., 2017*) and *Gata6* (*Schrode et al., 2014*). Alternatively, ICM cells might show increased sensitivity to the deregulation of such processes as RNA metabolism and cell cycle. Importantly, derepression of transposable elements (TE) might also result in increased DNA damage and cell death (*Ancelin et al., 2016*).

To address the latter hypothesis, we performed differential TE expression analysis on our RNA-seq dataset. Of particular interest are murine endogenous retrovirus-like elements (MuERV-L), which were shown to be derepressed upon loss of G9a in embryonic stem cells (ESCs)(*Maksakova et al., 2013*). This class of TEs is transiently expressed at the 2C stage (*Kigami et al., 2003*; *Macfarlan et al., 2012*; *Peaston et al., 2004*) and regulates zygotic genome activation, transcriptional networks and developmenal progression (*Huang et al., 2017*; *Kigami et al., 2003*). Our TE analysis of mutant embryos however revealed only a very mild reactivation of the ERV-L family of TEs, which contains MuERV-L (*Figure 3—figure supplement 3*). Thus, G9a is dispensable for timely repression of these retrotransposons. We have however observed persistent H3K9me2 staning at DAPI-dense foci in *Ehmt2$^{M/Z}$* at the 8C stage (*Figure 2A*), suggesting that another histone methyltransferase is likely to deposit silent chromatin marks at repetetive elements in the absence of G9a. Indeed, a recent study suggests that a loss of SETDB1, an H3K9 methyltransferase, results in upregulation of both ERV and LINE1 elements in oocytes (*Kim et al., 2016*).

In conclusion, the role of G9a during preimplantation development is distinct from that observed during postimplantation development. Notably, G9a is required for the repression of a specific set of transiently upregulated 4C genes in preimplantation embryos. Moreover, maternal G9a allows for the appropriate setting of transcriptional circuits, which promote developmental progression, timely PrE specification and stabilisation of ICM lineages. Together with PRDM14 and arginine methylation reported previously (*Burton et al., 2013*; *Torres-Padilla et al., 2007*), our findings suggest that the earliest epigenetic programming events at the onset of development are involved in creating a competent environment for cell fate choices that ensue. These epigenetic regulators, amongst others, ensure establishment of stable transcriptional circuitry, which instructs lineage segregation later in development.

# Materials and methods

**Key resources table**

| Reagent type (species) or resource | Designation | Source or reference |
| --- | --- | --- |
| Strain (mouse) | Ehmt2Flox/Flox | PMID:17707231 |
| Strain (mouse) | Zp3-Cre | PMID: 10686600 |

GO term enrichment analysis were performed exactly as in (*Zylicz et al., 2015*).

## Mouse breading, embryo collection

Timed natural matings were used for all experiments unless otherwise stated. Noon of the day when the vaginal plugs of mated females were identified was scored as E0.5. For *Ehmt2* matings a published conditional allele was used (*Sampath et al., 2007*). To obtain $Ehmt2^{Cntr}$ or $Ehmt2^{Mat}$ embryos, $Ehmt2^{F/+}$ Zp3-Cre$^{+ve}$ or $Ehmt2^{F/-}$ Zp3-Cre$^{+ve}$ females were used respectively (*de Vries et al., 2000*). When stated a *ΔPE-Pou5f1-EGFP* reporter line was crossed in (GGOF) (*Yeom et al., 1996*). All husbandry and experiments involving mice were carried out according to the local ethics committee and were performed in a facility designated by the Home Office.

## Immunofluorescence

Embryos were treated as previously described (*Nichols and Smith, 2009*). Primary antibodies used are as follows: anti-CDX2 (Biogenex, clone CDX2-88), anti-H3K9me2 (Abcam, UK, ab1220), anti-GFP (Nacalai tesque, Japan, GF090R), anti-G9a (Cell Signaling, MA, 68851T), anti-GLP (Research and Diagnostic Systems, MN, PP-B0422-00), anti-SOX2 (Abcam, UK, ab92494), anti-SOX17(Research and Diagnostic Systems, MN, AF1924). Mean nuclear intensities of IF and DAPI signal were quantified using mageJ and corrected for the staining background. As nuclear size is changing between stages all IF measurements were normalised to DAPI signal as a proxy for DNA content.

## Single-Embryo RNAseq

The embryos used were from natural matings and were morphologically assessed to ensure only viable samples were collected. cDNA was prepared and amplified as earlier described (*Tang et al., 2010*). Illumina libraries were prepared as published (*Huang et al., 2017*). Single-end 50 bp sequencing was performed with HiSeq4000 (Illumina, San Diego, CA). RNA-seq reads were adapter- and quality-trimmed, and aligned with Tophat2 (*Kim et al., 2013*) against the mouse reference (GRCm38/mm10) genome. Read counts per ENSEMBL transcript were obtained by SeqMonk. Differential expression was evaluated with the DESeq2 package (*Love et al., 2014*). Gene was deemed differentially expressed when p-value<0.05 after Benjamini and Hochberg correction. For clustering the dynamics of gene expression in distinct environments we have used an R package GeneCluster-Net based on Gaussian mixture fitting (*Wang et al., 2012*). For this analysis data was downloaded from GEO (GSE22182) (*Tang et al., 2011*). Optimal numbers of clusters (14) was identified by finding minimal Bayesian Information Criterion. For TE analysis, RNA-seq reads were aligned with *bowtie* (options: '-m 1 – v1 –best –strata') selecting for uniquely mapping reads only. RepeatMasker annotations of individual TE elements were downloaded from the UCSC Table Browser. Read counts per TE elements were obtained by *featureCounts* (http://bioinf.wehi.edu.au/featureCounts). Data is available under GSE106790.

# Acknowledgements

We are grateful to Alexander Tarakhovsky and Dónal O'Carroll for sharing G9a conditional knockout mice. We thank Dang Vinh Do for critical input into the project and members of the Surani Lab for helpful discussions.

## Additional information

### Funding

| Funder | Grant reference number | Author |
|---|---|---|
| Wellcome | 096738 | Jan J Zylicz<br>Maud Borensztein<br>Yun Huang<br>Caroline Lee<br>Sabine Dietmann<br>M Azim Surani |
| Wellcome | RG44593 | Jan J Zylicz |
| H2020 Marie Skłodowska-Curie Actions | 706144 | Maud Borensztein |
| Cancer Research UK | C6946/A14492 | Jan J Zylicz<br>Maud Borensztein<br>Yun Huang<br>Caroline Lee<br>Sabine Dietmann<br>M Azim Surani |
| James Baird Fund, University of Cambridge | | Yun Huang |
| Wellcome | 092096 | Jan J Zylicz<br>Maud Borensztein<br>Yun Huang<br>Caroline Lee<br>Sabine Dietmann<br>M Azim Surani |

The funders had no role in study design, data collection and interpretation, or the decision to submit the work for publication.

### Author contributions

Jan J Zylicz, Conceptualization, Data curation, Formal analysis, Validation, Investigation, Visualization, Methodology, Writing—original draft; Maud Borensztein, Validation, Investigation, Writing—review and editing; Frederick CK Wong, Validation, Investigation; Yun Huang, Sabine Dietmann, Resources, Data curation, Formal analysis, Investigation, Writing—review and editing; Caroline Lee, Investigation; M Azim Surani, Conceptualization, Supervision, Funding acquisition, Project administration, Writing—review and editing

### Author ORCIDs

Jan J Zylicz http://orcid.org/0000-0001-9622-5658
Maud Borensztein http://orcid.org/0000-0002-4378-5018
Yun Huang http://orcid.org/0000-0001-7843-9126
M Azim Surani http://orcid.org/0000-0002-8640-4318

### Ethics

Animal experimentation: Animal experimentation: All husbandry and experiments involving mice were authorised by a UK Home Office Project Licenses 80/2637 and PE596D1FE and carried out in a Home Office-designated facility.

### Decision letter and Author response

Decision letter https://doi.org/10.7554/eLife.33361.025
Author response https://doi.org/10.7554/eLife.33361.026

## Additional files

### Supplementary files
• Transparent reporting form
DOI: https://doi.org/10.7554/eLife.33361.014

### Data availability

Sequencing data have been deposited in GEO under accession codes GSE106790.

The following dataset was generated:

| Author(s) | Year | Dataset title | Dataset URL | Database, license, and accessibility information |
|---|---|---|---|---|
| Zylicz JJ, Borensztein M, Dietmann S, Huang Y, Lee C, Surani MA | 2017 | G9a regulates temporal preimplantation developmental program and lineage segregation in blastocyst | https://www.ncbi.nlm.nih.gov/geo/query/acc.cgi?acc=GSE106790 | Publicly available at the NCBI Gene Expression Omnibus (accession no: GSE106790). |

The following previously published datasets were used:

| Author(s) | Year | Dataset title | Dataset URL | Database, license, and accessibility information |
|---|---|---|---|---|
| Tang F, Barbacioru C, Lao K, Surani MA | 2010 | Global Deterministic and Stochastic Allelic Specific Gene Expression in Single Blastomeres of Mouse Early Embryos | https://www.ncbi.nlm.nih.gov/geo/query/acc.cgi?acc=GSE22182 | Publicly available at the NCBI Gene Expression Omnibus (accession no: GSE22182). |
| Wu J, Huang B, Chen H, Xie W | 2016 | The landscape of accessible chromatin in mammalian pre-implantation embryos (RNA-Seq) | https://www.ncbi.nlm.nih.gov/geo/query/acc.cgi?acc=GSE66582 | Publicly available at the NCBI Gene Expression Omnibus (accession no: GSE66582). |

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
