## [Decision Letter]

Thank you for submitting your article "G9a regulates temporal preimplantation developmental program and lineage segregation in blastocyst" for consideration by *eLife*. Your article has been favorably evaluated by Fiona Watt (Senior Editor) and two reviewers, one of whom, Asifa Akhtar (Reviewer #1), is a member of our Board of Reviewing Editors.

The reviewers have discussed the reviews with one another and the Reviewing Editor has drafted this decision to help you prepare a revised submission.

In this manuscript, authors study the functional consequence of the loss of maternally inherited G9a in blastocysts. Combination of immunofluorescence based image analysis and single embryo transcriptome analysis revealed that maternal loss of G9a results in misregulation of gene expression network leading to developmental delay. These data indicate that G9a mediated H3K9me2 is important for proper lineage commitment early in embryogenesis.

Although the reviewers find the manuscript interesting, it was discussed that the manuscript needs to be further strengthened by increasing the number of embryos analysed to be able to make more sound conclusions. In particular, the authors should address the following points upon revision:

- Authors conclude that development is unaffected up to the 8-cell stage, but by E4.5 defects emerge. However no analysis of developmental competence is presented. A careful examination of developmental progression in *Ehmt2^Mat^* embryos with observed embryonic stages on each day from E0.5 to E4.5 would be most appropriate to firmly establish when developmental defects arise.

In Figure 1, authors state that at least 3 embryos and 15 nuclei used for analysis. These are very low numbers (embryos) to form reliable conclusions. They should increase the N numbers to cover at least 3 independent experiments and at least 10 embryos per group.

- Immunostaining of H3K9me2 should also be performed at the 4-cell stage and morula stage to determine temporal dynamics of H3K9me2 in more detail. This is especially relevant in light of the results regarding derepression of 4-cell genes at the 8-cell stage in *Ehmt2^Mat^* embryos in Figure 3. The model proposed is that G9a is silencing genes between the 4-cell and 8-cell stage specifically (not before), which correlates with a global accumulation of H3K9me2. However this accumulation is only presented between the 2-cell and 8-cell stages here in Figure 1, not the 4-cell and 8-cell stages. If global accumulation of H3K9me2 is in fact occurring most prominently between the 2-cell and 4-cell stages the model proposed does not hold.

- Also in Figure 2, only 4 embryos analysed, which is not enough for an appropriate analysis. At least 3 independent experiments (and at least 10 embryos per group) are required.

- The authors conclude that there are fewer ICM cells in G9a-deficient blastocysts based on counting of Oct4-positive cells. However this conclusion is not fully supported by the results presented here. Please explain how this experiment was performed.

- It would also be most informative to analyse these counts at both E3.5 as well as E4.5 to determine the timing of the effect (i.e. is ICM establishment at E3.5 or differentiation into PE vs. EPI at E4.5 affected in the absence of G9a?).

- The authors state that the GO term most enriched in genes downregulated in *Ehmt2^Mat^* 8C embryos is chromatin silencing, which can be entirely explained by altered levels of H2A, MacroH2A and H2A.J transcripts. It is hard to believe that the strongest GO term enrichment with a highly significant p value can be entirely explained by 3 transcripts. Can the authors provide additional discussion/evidence for this?

- Authors should expand on their observations that there is increase DNA damage in the absence of maternal G9a. It was not clear why this is the case?

- Is there a way to experimentally address whether gene expression defects are a result of direct targeting of G9a to the promoters?

- It was interesting to see that G9a dispensable for transposon repression. Is it possible to know whether upon G9a loss, SETDB1 gets upregulated to compensate?

Other points:

1) Is the H3K9me2 antibody used specific for K9me2? Since most antibodies are known to cross-react with K9me3 and/or K9me1 – it will be important that the authors discuss/provide a statement and/or data towards this effect.

2) For the presentation of confocal images:

- The scale bars are missing in a number of images (e.g. Figure 1A and B).

- How was fluorescence intensity quantification carried out?

---

## [Author Response]

[…] Although the reviewers find the manuscript interesting, it was discussed that the manuscript needs to be further strengthened by increasing the number of embryos analysed to be able to make more sound conclusions. In particular, the authors should address the following points upon revision:- Authors conclude that development is unaffected up to the 8-cell stage, but by E4.5 defects emerge. However no analysis of developmental competence is presented. A careful examination of developmental progression in Ehmt2^Mat^ embryos with observed embryonic stages on each day from E0.5 to E4.5 would be most appropriate to firmly establish when developmental defects arise.

We address the point of developmental delay by detailed staging of E2.5 embryos, using 42 control embryos (6 litters) and 44 maternally depleted *Ehmt2* embryos (8 litters) from natural matings. The analysis indicated a slight developmental delay already at E2.5 (see Figure 2—figure supplement 1; subsection “G9a promotes developmental progression and primitive endoderm (PrE) segregation”, second paragraph and figure legend). Crucially, *Ehmt2^Mat^* embryos develop after 8C stage, and are morphologically indistinguishable from controls until morula stage, except this slight developmental delay. We have decided not to perform detailed staging at multiple developmental time points as it would require extensive number of mutant mice, which are currently beyond the scope of our animal facility capacities. Furthermore, we attempted to perform in vitro cultures of embryos from super-ovulated females throughout the preimplantation stages. However, from our preliminary experiments, we observed from 2C stage onwards, the phenotype of *Ehmt2^Mat^* becomes aggravated with multiple blastocysts developing entirely without the ICM. Therefore, we have concluded that extended in vitro culture would produce an artefactual phenotype.

In Figure 1, authors state that at least 3 embryos and 15 nuclei used for analysis. These are very low numbers (embryos) to form reliable conclusions. They should increase the N numbers to cover at least 3 independent experiments and at least 10 embryos per group.

We have now greatly increased both the cell and embryo numbers (see Author response table 1 below). The data is in Figure 1 and a new Figure 1—figure supplement 1.

G9aGLPH3K9me2NnNnNnE1.511201019917E2.51612311831387E4.5161601111012120

- Immunostaining of H3K9me2 should also be performed at the 4-cell stage and morula stage to determine temporal dynamics of H3K9me2 in more detail. This is especially relevant in light of the results regarding derepression of 4-cell genes at the 8-cell stage in Ehmt2^Mat^ embryos in Figure 3. The model proposed is that G9a is silencing genes between the 4-cell and 8-cell stage specifically (not before), which correlates with a global accumulation of H3K9me2. However this accumulation is only presented between the 2-cell and 8-cell stages here in Figure 1, not the 4-cell and 8-cell stages. If global accumulation of H3K9me2 is in fact occurring most prominently between the 2-cell and 4-cell stages the model proposed does not hold.

We address this point by extending our IF analysis for G9a and H3K9me2 to the 4-cell stage (E2.0; at least 10 embryos analysed). We have included our new results in Figure 1 and for clarity moved GLP IF staining into Figure 1—figure supplement 1. We show start of G9a accumulation at the 4C stage, and significant increase at the 8C stage. The H3K9me2 accumulation follows a broadly similar trend at the 4C and 8C stage albeit the strongest accumulation is observed already at 4C stage, with a significant gain at 8C, which could partially be due to newly-targeted regions. Our combined data together with RNAseq analysis indicates that the H3K9me2 accumulation at the 4C stage might not initially extend to the 4C-specific genes, which probably becomes established at the 8C stage when G9a levels increase. This likely explains the robust derepression of 4C genes at the 8C stage (see subsection “G9a and H3K9me2 accumulate at 4 and 8-cell stage”, last paragraph and subsection “G9a represses a subset of 4C upregulated genes”, fourth paragraph).

- Also in Figure 2, only 4 embryos analysed, which is not enough for an appropriate analysis. At least 3 independent experiments (and at least 10 embryos per group) are required.

We have now analysed more embryos (N=6, see Figure 2B). The analysis clearly shows reduced H3K9me2 staining, already reported in our initial manuscript. A similar assay was used reported recently in *eLife* used as few as 5 embryos per genotype (see: (Ancelin et al., 2016) – Figure 3.)

- The authors conclude that there are fewer ICM cells in G9a-deficient blastocysts based on counting of Oct4-positive cells. However this conclusion is not fully supported by the results presented here. Please explain how this experiment was performed.

We have used two different markers of pluripotency: Sox2 (Figure 2) and Oct4 GFP reporter (Figure 2—figure supplement 1). Note that Sox2 detects the pre-Epiblast at E4.5 and not the PrE (See: (Wicklow et al., 2014) – Figure 3C); on the other hand, Oct4 detects both the pre-Epi and PrE (See: (Wicklow et al., 2014) – Figure 6G). Thus, the presence of fewer Oct4+ve cells with a stable Sox2 positive cells is consistent with a loss of PrE cells, which is confirmed by analysis of the Sox17 positive PrE cells (Figure 2; see subsection “G9a promotes developmental progression and primitive endoderm (PrE) segregation” and Figure 2—figure supplement 1 legend).

- It would also be most informative to analyse these counts at both E3.5 as well as E4.5 to determine the timing of the effect (i.e. is ICM establishment at E3.5 or differentiation into PE vs. EPI at E4.5 affected in the absence of G9a?).

Our analysis thus far reveals relatively normal pre-Epiblast lineage specification, and defective delineation of the PrE at E4.5. The unaffected pre-Epiblast by necessity reflects normal specification of the ICM at E3.5, without which there would be neither pre-Epi nor PrE. In our opinion, the additional experiment asked by the reviewer is unlikely add enough further insight on the role of G9a in preimplantation development. Due to limited numbers of mutant animals and in accordance with the guidance in animal work (3Rs), we have decided not to perform the experiment.

- The authors state that the GO term most enriched in genes downregulated in Ehmt2^Mat^ 8C embryos is chromatin silencing, which can be entirely explained by altered levels of H2A, MacroH2A and H2A.J transcripts. It is hard to believe that the strongest GO term enrichment with a highly significant p value can be entirely explained by 3 transcripts. Can the authors provide additional discussion/evidence for this?

We thank the reviewers for pointing out this apparent contradiction. The confusion arises because the histone variants and canonical histones, are coded for by multiple loci. We have mapped only unique reads allowing us to quantify the expression of individual histone genes. In our dataset we have found downregulation of 18 histone genes coding for 3 histones (H2A, MacroH2A and H2A.J). All these genes belong to the Chromatin Silencing GO term (GO:0006342). Apart from these 18 genes only 2 others are also downregulated in mutant embryos and belong to this GO term (see detailed list of genes below). Thus, the observed 8.3 fold enrichment of this GO term is indeed almost entirely driven by histone upregulation. Additional confusion might come from the fact that in Figure 3—figure supplement 1C we have combined the expression of each histone variant originating from individual genes. We have now clarified this in the figure legend and subsection “G9a represses a subset of 4C upregulated genes”, second paragraph.

List of downregulated genes associated with chromatin silencing GO:

DOT1-like, histone H3 methyltransferase (*S.cerevisiae*)(Dot1l)H2A histone family, member J(H2afj)H2A histone family, member Y(H2afy)histone cluster 1, H2ab(Hist1h2ab)histone cluster 1, H2ac(Hist1h2ac)histone cluster 1, H2ad(Hist1h2ad)histone cluster 1, H2ae(Hist1h2ae)histone cluster 1, H2af(Hist1h2af)histone cluster 1, H2ag(Hist1h2ag)histone cluster 1, H2ah(Hist1h2ah)histone cluster 1, H2ai(Hist1h2ai)histone cluster 1, H2ak(Hist1h2ak)histone cluster 1, H2an(Hist1h2an)histone cluster 1, H2ao(Hist1h2ao)histone cluster 1, H2ap(Hist1h2ap)histone cluster 2, H2aa1(Hist2h2aa1)histone cluster 2, H2aa2(Hist2h2aa2)histone cluster 2, H2ab(Hist2h2ab)histone cluster 2, H2ac(Hist2h2ac)trinucleotide repeat containing 18(Tnrc18)

- Authors should expand on their observations that there is increase DNA damage in the absence of maternal G9a. It was not clear why this is the case?

Indeed, this is an intriguing question. It is worth noting that gH2Ax is not only a marker of increased DNA damage, because it often accumulates in embryonic stem cells even without genotoxic stress (Banath et al., 2009). gH2AX also marks cells at specific cell cycle stages (e.g. G2), as well as in cells with relatively open chromatin (Banath et al., 2009; Turinetto and Giachino, 2015). We have now included additional discussion of these points (subsection “G9a promotes developmental progression and primitive endoderm (PrE) segregation”, last paragraph).

- Is there a way to experimentally address whether gene expression defects are a result of direct targeting of G9a to the promoters?

There is no available dataset or G9a ChIPseq at E2.5, since no suitable protocols exists to perform such experiment on limited numbers of cells as in early embryos. Recent novel technologies allow for mapping the recruitment of some transcriptional factors in limited numbers of cells; however G9a is not suitable for such experiments as its binding is relatively dynamic and spreads over larger regions.

- It was interesting to see that G9a dispensable for transposon repression. Is it possible to know whether upon G9a loss, SETDB1 gets upregulated to compensate?

We have now explored this aspect in our RNAseq dataset. The results indicate that there is no significant upregulation of Setdb1 upon maternal loss of G9a (see Author response image 1). This implies that G9a activity is not targeted to TEs; it is likely that Setdb1 plays this role.

Other points:1) Is the H3K9me2 antibody used specific for K9me2? Since most antibodies are known to cross-react with K9me3 and/or K9me1 – it will be important that the authors discuss/provide a statement and/or data towards this effect.

The reviewers have pointed out that many of the histone-modification antibodies show substantial levels of cross reactivity. That is why we have used a monoclonal anti-H3K9me2 antibody (Abcam, ab1220), after validating it using dot-blot analysis (Active motif). There is no detectible cross-reactivity with either K9me1, or me3. Author response image 2 shows the specificity factor calculated using dot-blot quantification. Shown are top 10 marks recognised by this antibody, neither H3K9me1 nor me3 is showing detectable cross-reactivity.

**Author response image 2. respfig2:** 

2) For the presentation of confocal images:- The scale bars are missing in a number of images (e.g. Figure 1A and B).

The missing scale bars are now added.

- How was fluorescence intensity quantification carried out?

Mean nuclear intensities of IF and DAPI signal were quantified using the imaging software Image J and corrected for the staining background. As nuclear size changes between stages, all IF measurements were normalised to DAPI signal as a proxy for DNA content. We have now added this description in the Materials and methods subsection “Immunofluorescence”.